# Ultracompact Polarization Splitter–Rotator Based on Shallowly Etched Subwavelength Gratings and Anisotropic Metasurfaces

**DOI:** 10.3390/nano12193506

**Published:** 2022-10-07

**Authors:** Chengkun Dong, Sijie Dai, Jun Xia, Guodong Tong, Zhihai Wu, Hao Zhang, Bintao Du

**Affiliations:** Joint International Research Laboratory of Information Display and Visualization, School of Electronic Science and Engineering, Southeast University, Nanjing 210096, China

**Keywords:** silicon photonics, polarization splitter–rotator, anisotropic metasurfaces, subwavelength gratings

## Abstract

Polarization splitter–rotators (PSRs) are an essential component in on-chip polarization-sensitive and polarization–division multiplexing systems. In this work, we propose an ultracompact and high-performance silicon-based polarization splitter–rotator utilizing anisotropic metasurfaces, which is the first to combine the two, to our knowledge. The tilted periodic metasurface structure has different modulation effects on different polarized light fields, such as the transverse–electric (TE) mode and the transverse–magnetic (TM) mode, which are beneficial for designing polarization management devices. According to the results, the entire length of the silicon PSR was ~13.5 μm. The TE-to-TM conversion loss and polarization conversion ratio ere −0.154 dB and 96.5% at 1.55 μm, respectively. In the meanwhile, the cross talk and reflection loss were −27.0 dB and −37.3 dB, when the fundamental TE mode was input. The insertion loss and cross talk were −0.19 dB and −25.01 dB at the central wavelength when the fundamental TM mode was input. In addition, the bandwidth reached up to ~112 nm with polarization conversion loss and insertion loss higher than −0.46 dB and −0.36 dB. The simulations also show that the designed devices had good fabrication tolerance.

## 1. Introduction

The on-chip photonic integrated circuit (PIC) has received a lot of attention [1] from both academic and commercial communities because of its potential to overcome the challenges of traditional communication [2,3,4] and computing [5,6,7]. Among the many material platforms, silicon on insulator (SOI) [8,9,10] has resulted in extensive research on account of its large refractive index contrast, compatibility with complementary metal-oxide semiconductor (CMOS) technologies, and low optical absorption and loss in optical communication bands. However, the polarization-sensitive characteristics of silicon materials due to the inevitable birefringence effect [11] make both passive and active devices respond completely differently to transverse electric (TE) and transverse magnetic modes (TM). Thus the on-chip polarization control system was introduced [12].

As an essential polarization modulation device, the primary function of the polarization rotator (PR) [13] is to realize the interconversion between the fundamental TE mode and the fundamental TM mode. The polarization beam splitter–rotator (PSR) [14] is a device that realizes both polarization rotation and polarization beam splitting functions. Because of the large refractive index contrast between silicon and silica, the resulting birefringence makes the silicon waveguide highly capable of polarization maintenance, which makes it necessary to introduce some special asymmetric structures [15,16] to enable the hybrid coupling of the two modes of TE and TM. In ref. [17], the waveguide realized the function of the PR based on adiabatic transition conditions using a width gradient etched structure. Ref. [18], on the other hand, achieved polarization rotation in a length of about 22.1 μm by uniformly etching the structure. Xiong et al. [19] presented a PSR based on a directional coupler with a taper-etched width, which could reach a 30 dB extinction ratio bandwidth of 160 nm, but the overall size was more than 80 μm. In addition, waveguide structures using air cladding or SiN cladding [20] are also available. However, it remains a challenge to achieve ultra-compact on-chip polarization management devices with superior performance that require low transmission loss (EL) and a high extinction ratio (ER) over an extensive spectral range. The integration of optical waveguides and metasurface structures is promising to help improve device performance [21].

Metasurfaces, which are two-dimensional artificial electromagnetic materials [22,23,24] with subwavelength features, can be used to control the phase, amplitude, polarization, and optical impedance of light. Many integrated optical devices have been realized utilizing metasurface structures, e.g., mode converters [25,26] and polarization beam splitters [27,28,29,30] (also referred to as subwavelength grating structures in some papers). More recently, anisotropic metasurface (AMS) structures [31,32] have also attracted increasing research interest because of their different responses to different polarizations. In ref. [33], a PBS based on AMS structure is presented, and it realized an extinction ratio of 19.84 dB and a coupling length of 6.8 μm. Ref. [21] utilized tilted plasmonic metasurfaces to construct PR at a central wavelength of 4 um. However, the mode purity was relatively small.

In this paper, a compact and high-performance PSR utilizing shallowly etched SWGs and AMS structures is proposed. There are usually two ways to achieve mode rotation: shallow etching and changing the upper cladding material [34]. Because the polarization rotation efficiency is low by changing the upper cladding material, the shallow etching scheme was adopted in this study. For the input TE mode, the mode field distributions spread throughout the AMS structures and the SWG structures in turn, and finally rotate into the TM mode at the cross-port output, which is based on the adiabatic transition mechanism. On the contrary, the TM mode is confined to the input strip waveguide away from the polarization rotation region and eventually output at the bar port. Section 2 of the research paper describes the structural components and operating principles of the device. Section 3 analyzes the impact of several key parameters on device performance and optimizes them based on the results. Section 4 analyzes the spectral response and fabrication tolerance of the proposed PSR.

## 2. Device Structure and Principle

Figure 1 presents a three-dimensional and two-dimensional schematic of the PSR, including a side view and a top view of the mode evolution region. The proposed PSR contains subwavelength grating structures and anisotropic metasurface structures. The entire SWG structure has the same period Λ_1_ and duty cycle a_1_/Λ_1_, where a_1_ is the width of the silicon in the grating. The subwavelength grating region is partially etched in order to break the mode symmetry in the waveguide, which facilitates mode rotating. The height of the partially etched region is H1, and the rest has a height of H_2_, as shown in Figure 1b. The period and duty cycle of the AMS structure are Λ_2_ and a_2_/Λ_2_, respectively.

The TM mode has the primary component of the electric field along the y-direction and is, therefore, less affected by the tilt angle, while the TE mode is more affected, as will be discussed later. The device consists of the following main parts: the narrow strip waveguide, the input transition region (regions L_1_), the coupling region (region L_2_), the two output transition regions (regions L_3_ and L_4_), and the two output ports.

The width of the input strip waveguide is W_1_, and then in the input transition region L_1_, the width of the strip waveguide gradually changes to W_2_. At the same time, the width of the AMS structure gradually increases from 0 to W_6_, where the angle of each meta-atom to the X-axis is *θ*. Immediately following the input transition region is the mode evolution region. In this region, the width of the strip waveguide W_2_ and the width of the anisotropic metasurface structure W_6_ remains constant, while the width of the output strip waveguide W_3_ remains constant for a certain distance and gradually changes to W_4_. The output transition region follows the mode evolution region. The output transition region of the bar port is the region L_3_, where the width of the meta-atoms decreases to 100 nm and is spatially away from the SWG structure. The output transition region for the cross port is region L_4_, where the width of the SWG is gradually reduced to 100 nm.

When the TE mode is input, the mode field first passes through the transition region L_1_. Since the striped narrow waveguide does not support the TE mode well, its mode field distribution gradually spreads throughout the AMS structure in this region. In the mode evolution region, the SWG structure has a higher equivalent refractive index to the TE mode compared to the AMS structure. Therefore, the TE mode spreads to the SWG region and is further coupled to the output strip waveguide. In the meanwhile, due to the partially etched SWG and waveguide forming an asymmetric structure, a polarization rotation effect occurs, and the TE mode will rotate to the TM mode. The converted TM mode outputs from the cross port.

When the TM mode is input, its mode field can be tightly confined in the strip narrow waveguide with width W_2_, and only a tiny portion of energy will leak into the AMS structure, which is precisely the opposite of the mode field distribution of the TE mode. Since the guided TM mode in the taper region will be far away from the SWG region, it is less likely to couple to the output strip waveguide. The AMS structure will have a lower equivalent refractive index for the TM mode as a result of the presence of the meta-atomic tilt angle (although the difference is not as large as for the TE mode), which allows the TM mode to be better confined within the strip waveguide, further improving the extinction ratio of the TM mode. The TM mode is finally exported from the bar port. Since both the TE and TM modes are input from the input strip waveguide, and only the TM mode is required to be output from the bar port, W_1_ is slightly larger than W_2_. Combined with the above description of the device principle, light can only be input from the W_1_ port. If the input is the TE mode, it will rotate into the TM mode and output from the cross port; if the input is the TM mode, it will output directly from the bar port.

Figure 2a,b show the mode field distributions of the TM mode and TE mode in the input strip waveguide, respectively. The polarization conversion efficiency (PCE) [18] can be expressed as follows:(1)$$PCE=\sin^{2}\left(2\theta \right)\sin^{2}\left(\frac{\pi L}{2L_{c}}\right)\times 100\%$$(2)$$\theta =\tan^{-1}\left(\frac{\iint n^{2}\left(x,y\right)H_{x}^{2}\left(x,y\right)dxdy}{\iint n^{2}\left(x,y\right)H_{y}^{2}\left(x,y\right)dxdy}\right)$$
where $L_{c}$ is the polarization rotation length, *L* is the entire length of the mode conversion section, θ is the optical axis rotation angle, n(x,y) is the refractive index distribution, $H_{x}\left(x,y\right)$ is the transverse components of an eigenmode, and $H_{y}\left(x,y\right)$ is the horizontal components of an eigenmode. It can be seen from the above equation that the optical axis must be deflected at an angle of 45° to achieve 100% polarization conversion efficiency, as shown in Figure 2c,d. By the designed asymmetric structure, the two fundamental modes TE and TM are almost fully hybridized, with a large overlap of field components, which leads to a high power conversion efficiency.

Some of the structural parameters of this PSR are as follows: the relative refractive indices of silicon and silica were taken as 3.478 and 1.444, respectively (at 1.55 μm); the height of the partially etched area was 170 nm and the rest was 340 nm. Below the silicon layer was the oxide buried layer with a thickness of 2 μm, and the cladding layer was also 2 μm thick.

## 3. Simulations and Discussion

To research the transmission characteristics of the device in detail, we used the polarization conversion loss, polarization conversion ratio, crosstalk, insertion loss, and reflection loss to characterize the properties of the PSR.

For the TE mode, the polarization conversion loss (PCL), polarization conversion ratio (PCR), crosstalk (CT), and reflection loss (RL) are calculated as below:(3)$$\mathrm{PCL}\left(\mathrm{dB}\right)=10\log_{10}\left(\frac{P_{\mathrm{TM}}^{\mathrm{CRO}}}{P_{\mathrm{TE}}^{\mathrm{In}}}\right)$$
(4)$$\mathrm{PCR}\left(\%\right)=\frac{P_{\mathrm{TM}}^{\mathrm{CRO}}}{P_{\mathrm{TE}}^{\mathrm{In}}}\times 100\%$$
(5)$$\mathrm{CT}\left(\mathrm{dB}\right)=10\log_{10}\left(\frac{P_{\mathrm{TE}}^{\mathrm{Bar}}}{P_{\mathrm{TM}}^{\mathrm{CRO}}}\right)$$
(6)$$\mathrm{RL}\left(\mathrm{dB}\right)=10\log_{10}\left(\frac{P_{\mathrm{TE}}^{R}}{P_{\mathrm{TE}}^{\mathrm{In}}}\right)$$

For the TM mode, the insertion loss (IL), crosstalk (CT), and reflection loss (RL) are calculated as below:(7)$$\mathrm{IL}\left(\mathrm{dB}\right)=10\log_{10}\left(\frac{P_{\mathrm{TM}}^{\mathrm{Bar}}}{P_{\mathrm{TM}}^{\mathrm{In}}}\right)$$
(8)$$\mathrm{CT}\left(\mathrm{dB}\right)=10\log_{10}\left(\frac{P_{\mathrm{TM}}^{\mathrm{CRO}}}{P_{\mathrm{TM}}^{\mathrm{In}}}\right)$$
(9)$$\mathrm{RL}\left(\mathrm{dB}\right)=10\log_{10}\left(\frac{P_{\mathrm{TM}}^{R}}{P_{\mathrm{TM}}^{\mathrm{In}}}\right)$$
where $P_{\alpha }^{\beta }$ represents the optical power of the α mode (the fundamental TE or TM mode) at the β port (In, input port; R, reflection port; Bar, bar port; CRO, cross port). In the next numerical simulations, the incident optical power was generally normalized to 1, and the central operating wavelength was fixed at 1550 nm. The initial parameters of the device were as follows: the width of the input strip waveguide W_1_ was 0.32 μm, the length of the mode evolution region L_2_ was 5.5 μm, the period of SWG $\Lambda_{1}$ was 0.23 μm, the width of the silicon in the grating a_1_ was 0.13 μm, the width of the SWG structure W_5_ was 0.4 um, and the width of the AMS structure W6 was 0.52 μm. The PSR was analyzed and optimized by the three-dimensional finite-difference time-domain (3D-FDTD) method [35].

It can be seen in Figure 1 that Λ_1_ has the following relationship with Λ_2_
(10)$$\Lambda_{2}=\Lambda_{1}\times \cos\theta$$

If the metasurface structure works in the deep-sub-wavelength regime, it can be regarded as an equivalent homogeneous material, which has the following refractive index according to Rytov’s formulas [36]:(11)$$n_{∥}^{2}=\frac{a}{\Lambda_{1}\times \cos\theta }n_{1}^{2}+\left(1-\frac{a}{\Lambda_{1}\times \cos\theta }\right)n_{2}^{2}$$
(12)$$\frac{1}{n_{⊥}^{2}}=\frac{a}{\Lambda_{1}\times \cos\theta }\frac{1}{n_{1}^{2}}+\left(1-\frac{a}{\Lambda_{1}\times \cos\theta }\right)\frac{1}{n_{2}^{2}}$$
where $n_{∥}$ and $n_{⊥}$ represent the refractive indices in the parallel and perpendicular directions of polarization, respectively. The tilt angle *θ* of the meta-atom leads to a rotation of the diagonal permittivity tensor $\tilde{\varepsilon }$ [32]:(13)$$\tilde{\varepsilon }\left(\theta \right)=T^{-1}\left(\theta \right)\varepsilon T\left(\theta \right)=\left[\begin{array}{ccc} {\tilde{\varepsilon }}_{xx} & 0 & {\tilde{\varepsilon }}_{xz}\\ 0 & {\tilde{\varepsilon }}_{yy} & 0\\ {\tilde{\varepsilon }}_{xz} & 0 & {\tilde{\varepsilon }}_{zz} \end{array}\right]$$
(14)$$\{\begin{array}{c} {\tilde{\varepsilon }}_{xx}\left(\theta \right)=n_{∥}^{2}\cos^{2}\left(\theta \right)+n_{⊥}^{2}\sin^{2}\left(\theta \right)\\ {\tilde{\varepsilon }}_{yy}\left(\theta \right)=n_{∥}^{2}\\ {\tilde{\varepsilon }}_{zz}\left(\theta \right)=n_{∥}^{2}\sin^{2}\left(\theta \right)+n_{⊥}^{2}\cos^{2}\left(\theta \right)\\ {\tilde{\varepsilon }}_{xz}\left(\theta \right)=(n_{⊥}^{2}-n_{∥}^{2})cos\left(\theta \right)sin\left(\theta \right) \end{array}$$
where T is the rotation matrix in the above equation. Figure 3a gives the values of ${\tilde{\varepsilon }}_{xx}$ and ${\tilde{\varepsilon }}_{yy}$ as θ varied at the fixed duty cycles of 35%, 45%, and 55%. In the range of 0°–36° for θ, ${\tilde{\varepsilon }}_{xx}$ decreased as θ increased, while ${\tilde{\varepsilon }}_{yy}$ was less affected by θ. Since the effective refractive index of the TE mode and TM mode were mainly influenced by ${\tilde{\varepsilon }}_{xx}$ and ${\tilde{\varepsilon }}_{yy}$, respectively, the effective refractive index of the TE mode decreased with the increase in θ, and the effective refractive index of the TM mode was less influenced by θ. This also explains why, for the input TE mode, the performance of the PSR was more affected by the tilt angle, while when inputting the TM mode, the performance was less affected, as can be seen in Figure 3b,c. As theta increased, the effective refractive index of the TE mode in the AMS structure decreased, and thus, more energy was distributed into the output strip waveguide, which in turn, caused an increase in the polarization conversion ratio of the TE mode. When theta was > 20 degrees, the polarization conversion ratio of the TE mode decreased again. This is because the equivalent refractive index of the TE mode decreased further, the binding of the TE mode by the input strip waveguide increased, more energy remained in the input strip waveguide, and the polarization conversion ratio of the TE mode decreased. Therefore, the chosen angle was 17°–19°. Within this range, the PCL and IL of the PSR were greater than −0.3 dB, while the CT was less than −21 dB.

Figure 4a,b shows the transmission performance of the PSR as a function of the grating period Λ_1_ of the SWG structure. In order to operate in the subwavelength region, the period of the SWG structure has to be much smaller than $\Lambda_{Bragg}=\Lambda /\left(2n_{eff}\right)$. Therefore, the period range of our simulation was 0.20 μm–0.28 μm. It is worth noting that the polarization conversion ratio of the TE mode shows an increasing trend when Λ_1_ < 230 nm. On the one hand, this was because when Λ_1_ decreased, the duty cycle of the SWG structure increased and the equivalent refractive index to the TE mode increased, which may have excited a higher-order TE1 mode, and thus, caused a decrease in the polarization conversion ratio. On the other hand, it can be seen in Equation (10) that an increase in Λ_1_ resulted in a corresponding increase in Λ_2_, which reduced the equivalent refractive index of the AMS structure for the TE mode, and facilitated the diffusion of the TE mode into the SWG structure.

When Λ_1_ > 230 nm, the effective refractive index of TE mode further decreased, which then left some energy in the strip waveguide, decreasing the polarization conversion ratio of the TE mode. Figure 4b shows that the TM mode was insensitive to Λ_1_. The inappropriate value of Λ_1_ could also result in a deflection angle that is not 45°. According to the results shown in Figure 4a, the optimum value of Λ_1_ was 230 nm, and the corresponding values of PCL, CT, RL, and PCR were −0.21 dB, −25.62 dB, −34 dB, and 95.07% for the TE mode and those of the IL, CT, and RL were −0.22 dB, −22.83 dB, and −37.46 dB for the TM mode.

The next parameters to be determined were the duty cycle of the SWG structure and the AMS structure. The polarization conversion loss and insertion loss were calculated for the TE mode and the TM mode at a wavelength of 1550 nm, with a_1_/Λ_1_ ranging from 0.48 to 0.60 and a_2_/Λ_2_ ranging from 0.32 to 0.47. It can be seen in Figure 5b that the smaller the a_2_/Λ_2_ was, the higher the IL of the input TM mode was. This was because the smaller the a_2_/Λ_2_, the less effective refractive index the AMSs region had for the fundamental TM mode, which was transmitted to the input strip waveguide with little leakage. In Figure 5a, the PCL of the input TE mode is correlated with both a_1_/Λ_1_ and a_2_/Λ_2_. The improper duty cycle combination made more energy leak into the AMS structure and eventually radiated into the cladding or retain in the strip waveguide, which caused a decrease in performance. When the duty cycle was taken in the range of the white dashed box, the device had a better performance. From the perspective of device manufacturing, here, we took a_1_/Λ_1_ as 0.565 and a_2_/Λ_2_ as 0.4113 (a_1_ is 0.13, a_2_ is 0.09), and both the TE mode and the TM mode had high polarization conversion efficiency and insertion loss of −0.217 and −0.218, respectively.

Figure 6 shows the performance of the PSR at wavelengths of 1.45 μm–1.65 μm for different lengths of L_2_. For the input TE mode, the central wavelength of the CT gradually shifted toward 1.55 μm, and the conversion efficiency of the PSR gradually increased when L_2_ was greater than 5.4 um. When L_2_ was larger than 5.7 um, the polarization conversion efficiency of the PSR at long wavelengths decreased gradually with the increase in L_2_. For the input TM mode, there was a slight increase in the IL of the device as L_2_ increased.

The optimal waveguide structure parameters were determined as θ = 18°, Λ_1_ = 0.23 μm, a_1_/Λ_1_ = 0.565, a_2_/Λ_2_ = 0.4113, and L_2_ = 5.7 μm, and the light propagation properties for the TE and TM mode at the central wavelength are presented in Figure 7. Figure 7a,b,d shows the case of the input TE mode and Figure 7c,e shows the case of the input TM mode. Ex is the main component of the TE mode and Ey is the main component of the TM mode. In the device, the TE mode was gradually coupled from the input strip waveguide to the AMS structure and eventually rotated into the TM mode output at the cross port. The TM mode was transmitted along the input strip waveguide and output at the bar port.

In order to further improve the extinction ratio of the device, a polarization filter structure was added to the cross and bar ports. The specific structure was a strip waveguide with a gradual change in width to 120 nm, where the TE mode was cut off and gradually dissipated into the cladding from the conduction mode to the radiation mode. With the addition of polarization filters, the total device length increased by 1.5 μm [37].

## 4. Device Performance and Fabrication Tolerance Analysis

The wavelength-dependent characteristics of the device transmission performance were analyzed for the wavelength range of 1450 nm to 1650 nm (covering the S band, C band, L band, and a part of the U band). The transmission performance of the TE mode and the TM mode with tilt angles of 0° and 18° are shown in Figure 8a,b, respectively.

For the input TM mode, the IL of the PSR was greater than 0.62 dB, the CT was less than −22 dB, and the RL was less than −25 dB over the entire band. For the input TE mode, the PCR of the PSR was greater than 80% from 1451 nm to 1622 nm. Within the wavelength range, CT and RL were less than −17 dB and −26 dB, respectively. The optimal values of the PCL, CT, and RL were reached near the central wavelength of 1550 nm for the input TE mode, while for the input TM mode, the performance of the PSR decreased with longer wavelengths. This was due to the fact that when the wavelength was short, some energy remained in the strip waveguide, which was detrimental to the input TE mode. At longer wavelengths, the AMS structure enhanced the attraction to the TE and TM modes, thus leading to reduced device performance. The device was significantly better at a tilt angle of 18° than at 0° due to the formation of an effective refractive index gradient that facilitated the diffusion of the TE mode into the cross port (n_TE_ > n_eff2_TE_ > n_eff1_TE_), as shown in Figure 1a.

Finally, the manufacturing tolerances of the device were analyzed. According to the above description, the fabrication tolerances of a_1_, a_2_, L_2_, theta, etc., were initially analyzed, so the tolerance analysis of the other key parameters, such as W_5_, W_6_, H_1_, n_buried layer_, and n_cladding layer_ are given below. The subwavelength grating region is closely connected to the anisotropic metasurface (AMS) region. When the length of the subwavelength grating changed, the shape and position of the AMS structure also moved accordingly, and the tolerance is discussed here. In Figure 9, it can be found that ΔW_5_ and ΔW_6_ had little influence on the device performance for the TM mode. As the values of ΔW_5_ and ΔW_6_ became larger, the TM mode gradually moved away from the cross bar, which caused the IL to increase. When ΔW_5_ and ΔW_6_ were in the ranges of (−18 nm, 30 nm) and (−10 nm, 30 nm), the CT was less than −25 dB for the TE mode. In order to keep the PCL within −0.6 dB, the ΔH1 must be controlled from (−15 nm, 12 nm). Due to the process, the refractive indices of the buried and cladding layer were different. As can be seen in the figure, n_cladding layer_ had a slightly greater effect on the device than n_buried layer_, because n_cladding layer_ affected the optical axis deflection angle. When n_buried layer_ and n_cladding layer_ were in the ranges of (1.41, 1.47) and (1.43, 1.455), the PCL and IL were better than −0.2 dB and −0.25 dB.

The manufacturing error can be kept within 10 nm [38] under the conditions of the existing manufacturing process. The proposed PSR is able to be readily realized by two-step deep-ultraviolet lithography or an electron-beam lithography (EBL) process and reactive ion etching process. Table 1 summarizes the performance of PSRs utilizing various structures. The proposed PSR has a theoretically better performance compared to them.

## 5. Conclusions

We have designed a high-performance and ultracompact silicon-based PSR based on the partially etched SWG and AMS structures. By introducing anisotropic metasurfaces that respond differently to different polarization, the freedom of the device design was increased. The lower effective refractive index of the AMS structure for the TE mode made it easier to couple it to the SWG structure and eventually rotate it into the TM mode output. For the input TM mode, the AMS structure was equivalent to its cladding layer to keep it away from the shallow etched area to avoid mode rotation. With the optimized structure, the length of the whole device was about 13.5 μm. At a wavelength of 1550 nm, the PCL for the TE mode and the IL for the TM mode were −0.154 dB and −0.19 dB, respectively, with a low CT (−27.0 dB/−25.01 dB) and a low RL (−37.3 dB/−37.2 dB). The bandwidth can be enlarged to 112 nm with PCL < −0.46 dB and IL < −0.36 dB.

## Figures and Tables

**Figure 1 nanomaterials-12-03506-f001:**
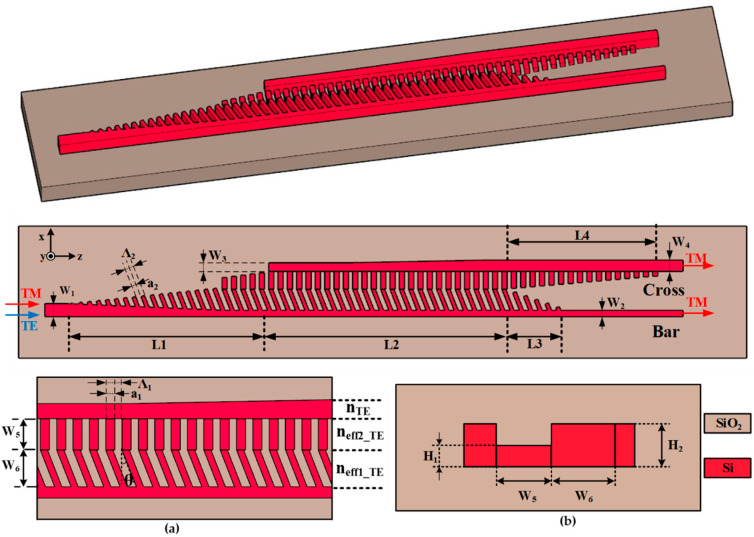
The structure of the device: (**a**) partially enlarged view and (**b**) cross-sectional view.

**Figure 2 nanomaterials-12-03506-f002:**
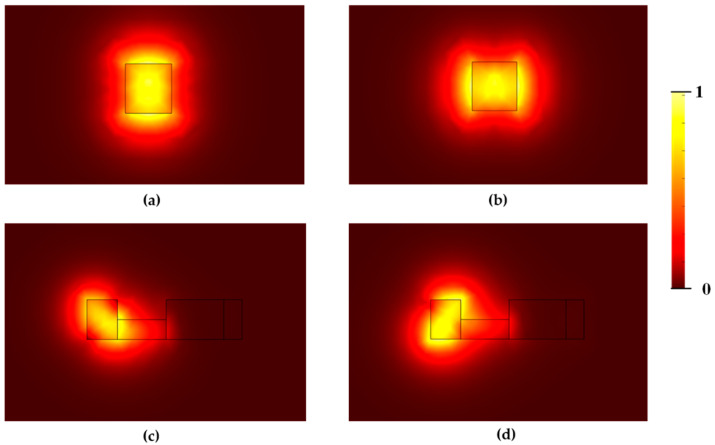
Mode field distributions of the fundamental (**a**) TM and (**b**) TE modes supported by the input strip waveguide. (**c**,**d**) Optical axis is deflected by 45° at the middle of L_2_.

**Figure 3 nanomaterials-12-03506-f003:**
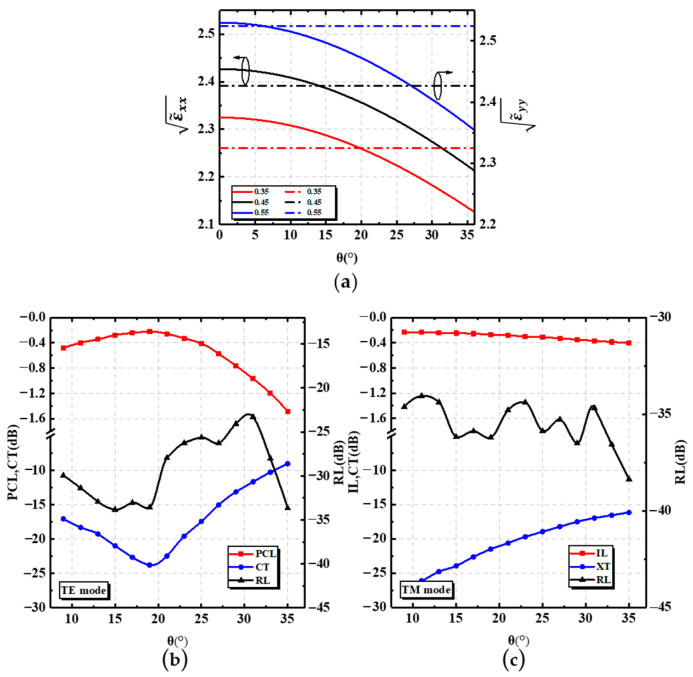
(**a**) $\sqrt{{\tilde{\varepsilon }}_{xx}}$ and $\sqrt{{\tilde{\varepsilon }}_{yy}}$ as functions of θ. Dependence of PCL, CT, and RL on θ for (**b**) the TE mode and for (**c**) the TM mode.

**Figure 4 nanomaterials-12-03506-f004:**
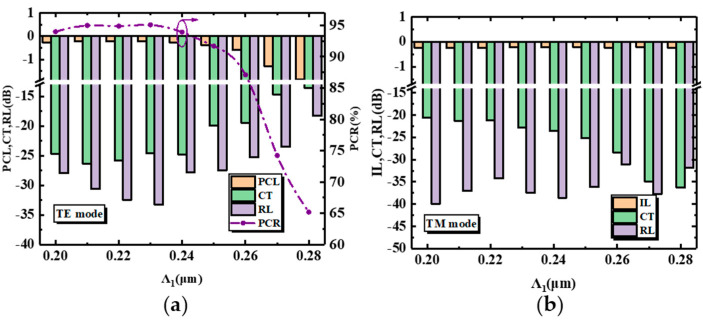
(**a**) PCL, CT, RL, and PCR of the proposed PSR versus Λ_1_ for the TE mode. (**b**) IL, CT, and RL of the proposed PSR versus Λ_1_ for the TM mode.

**Figure 5 nanomaterials-12-03506-f005:**
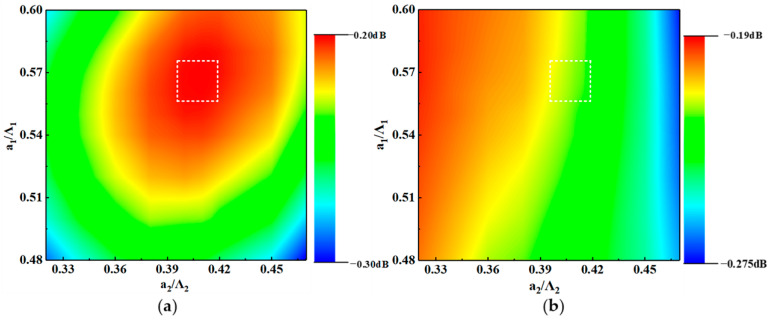
(**a**) PCL and (**b**) IL of the PSR for the TE mode and TM mode with different a_1_/Λ_1_ and a_2_/Λ_2_. When the value of the duty cycle was within the white dashed box, the overall performance of the device was better.

**Figure 6 nanomaterials-12-03506-f006:**
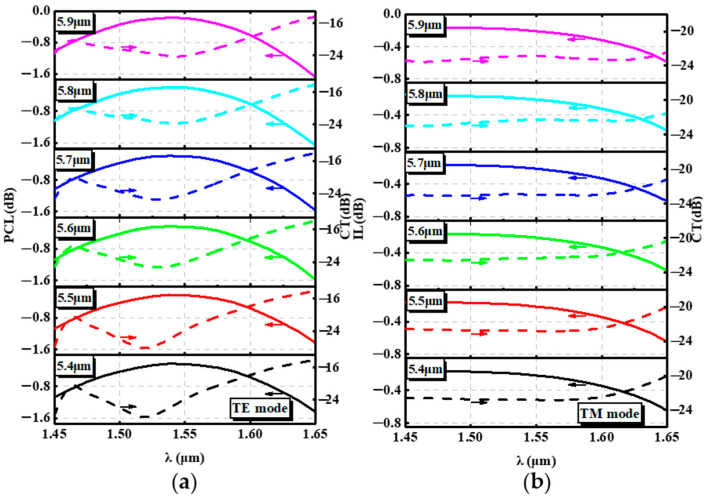
(**a**) PCL and CT of the PSR with different L_2_ from 1.45 μm to 1.65 μm. (**b**) IL and CT of the PSR with different L_2_ from 1.45 μm to 1.65 μm.

**Figure 7 nanomaterials-12-03506-f007:**
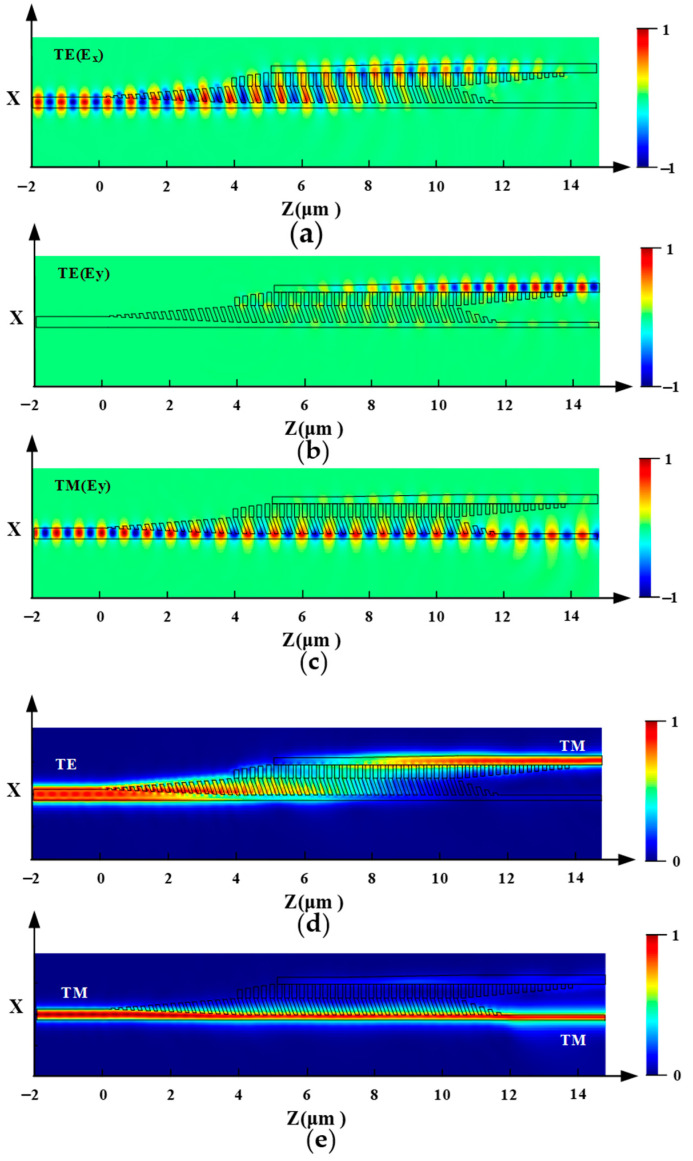
Mode evolution of (**a**) the E_x_ and (**b**) E_y_ for the fundamental TE mode, and (**c**) E_y_ for the fundamental TM mode with the optimal parameter. Light intensity of (**d**) the input TE and (**e**) TM modes through the proposed PSR.

**Figure 8 nanomaterials-12-03506-f008:**
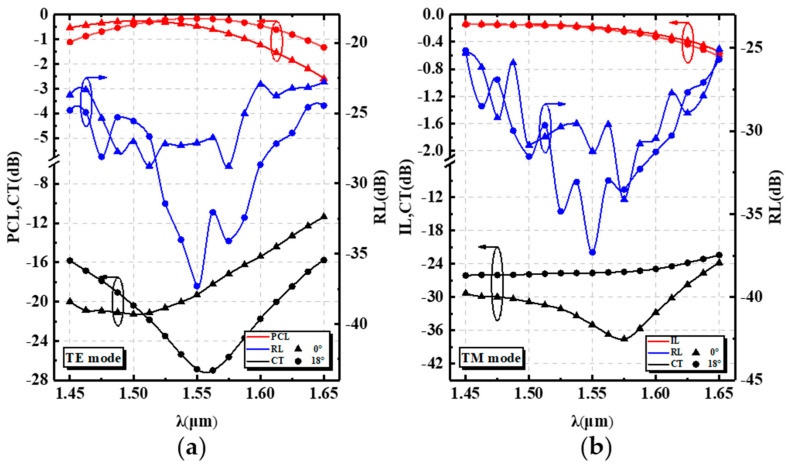
Comparison of the proposed device properties as functions of the wavelength at tilt angles of 0° and 18° for (**a**) the input TE mode and (**b**) TM mode.

**Figure 9 nanomaterials-12-03506-f009:**
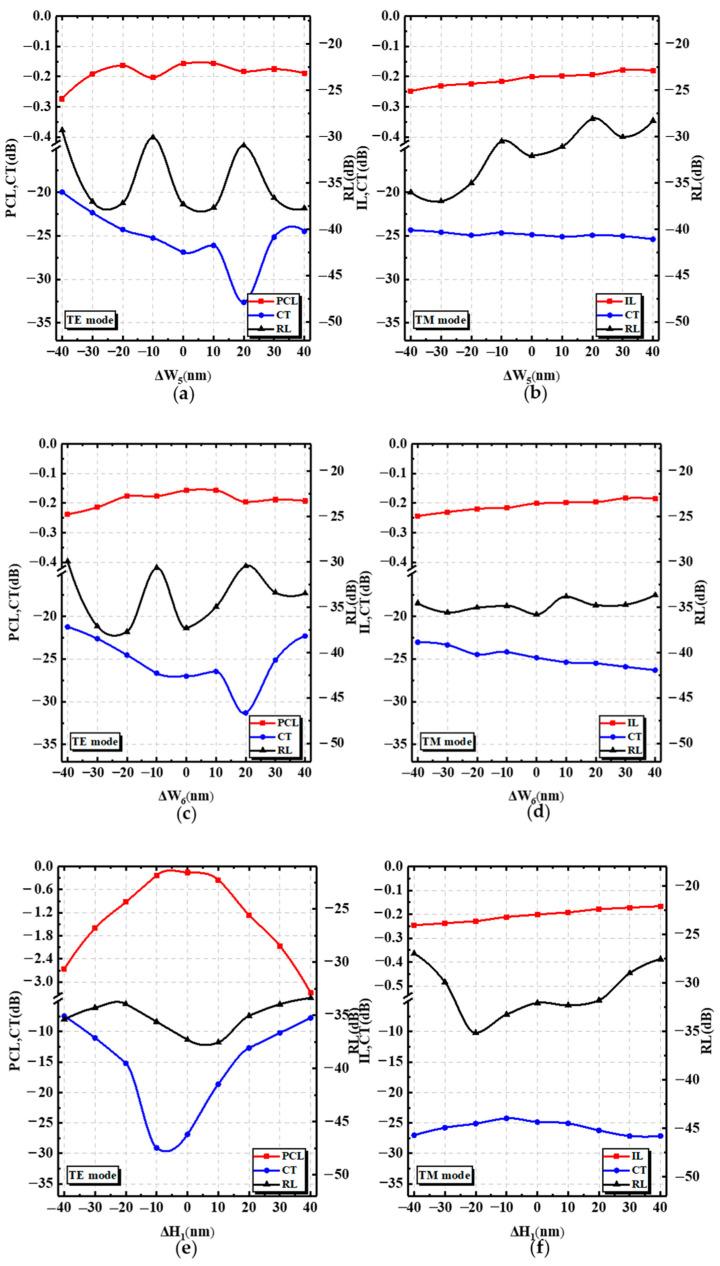
The analysis of the fabrication tolerance. (**a**,**c**,**e**,**g**,**i**) show the PCL, CT, and RL of the PSR with different W_5_, W_6_, H_1_, n_buried layer_, and n_cladding layer_ deviations for the TE mode. (**b**,**d**,**f**,**h**,**j**) present the IL, CT, and RL with different W_5_, W_6_, and H_1_, n_buried layer_, and n_cladding layer_ deviations for the TM mode.

**Table 1 nanomaterials-12-03506-t001:** Comparison of the reported PSR.

Structures	Footprint	PCL	PCR	ER	Bandwidth	Fabrication
PSR based on SWG [39]	50 μm	−0.13 dB	97%	10 dB	35 nm (PCL > −0.4 dB)	No
PSR based on asymmetrical directional coupler [40]	28 μm	−0.18 dB	95.9%	28 dB	45 nm (PCL > −1 dB)	Yes
PSR based on slanted silicon waveguides [41]	55 μm	/		19.92 dB	100 nm (PCL > −0.5 dB)	No
PSR based on rib directional coupler [42]	24 μm	−0.13 dB	97%	/	100 nm (ER < 19 dB)	No
This work	~13.5 μm	−0.15 dB	96.5%	20.34 dB	112 nm (PCL > −0.46 dB)	No

## Data Availability

The data presented in this article are available upon request from the corresponding author.

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
