# Peer review of "Ultracompact Polarization Splitter–Rotator Based on Shallowly Etched Subwavelength Gratings and Anisotropic Metasurfaces"

_nanomaterials, 2022, doi:10.3390/nano12193506_

Round 1

Reviewer 1 Report

The authors report a polarization rotator from TE to TM from simulation results. 

They assume shallowly etched subwavelength grating of Si. 

Numerically, the polarization conversion is shown at C band. The efficiency 

is fairly high. Also, tolerance is studied regarding a few structural parameters. 

The design is explained in terms of effective media. 

Although the manuscript is basically comprehensive, there are several unclear 

points as follows. 

1) Figure 1

W1, W3 are hard to see, which should be improved. 

It is not clear why W1 and W2 are different in dimension and how W3 is determined. 

2) General

TE-to-TM conversion is shown. 

Does this structure work as TM-to-TE conversion device? 

If TM input is given at W3, then is TE output gotten at W1? 

At least, the authors need to describe it. 

3) Figure 2

The positions of the mode distribution are not described in the caption. 

Z positions should be specified. 

The distribution seems to be intensity or absolute value of electric field. 

However, the scale bar looks like indicating snapshot of electric field. 

It should be described explicitly. 

Also, electric-field vectors are helpful for readers to confirm polarizations 

in each panel. 

4) Figure 5

White broken-line boxes should be described in the caption. 

5) Figure 7

Regarding (c)-(e), polarizations are comprehensive. 

However, notations in (a) and (b) are confusing. 

(a) TE(Ex) seems to a mistake, which should be TE(Ey) in the coordinate. 

(b) Is TE(Ey) a mistake of TM(Ey)? 

6) Figure 9

What happens if W5 and W6 change together? 

In fabrication, W5 change will affect W6. They are not independent. 

7) General

Shallowly etched structures are difficult in reality. 

Readers will want to know whether or not it is possible to design polarization 

conversion device at the same height. This point should be discussed.

Reviewer 2 Report

Referee report

 In the paper entitled "Ultracompact Polarization Splitter-Rotator Based on the Shallowly Etched Subwavelength Gratings and Anisotropic Metasurfaces”, Chengkun Dong et al propose a silicon-based polarization splitter-rotator based on the anisotropic metasurfaces. Authors demonstrate that the by utilizing the shallowly etched slot waveguide one can achieve the TE-TM conversion at a remarkably high efficiency and compact side.

 My feeling is that the paper deserves publishing and may be interesting for the Materials readership. I suggest authors to address the two issues prior to publication:

1)    An extensive numerical analysis of the performance and fabrication tolerance of the proposed device employs the bulk refraction indexes of the materials (Si/SiO2) involved (page 3). However, it is not always the case. Few microns thick dielectric layers created by physical deposition may have refraction indices essentially different from those in the bulk materials. As a result, such parameters as the bandwidth may vary in a wide range. I suggest authors to evaluate the fabrication tolerance in terms of refractive indices of the buried and cladding layers.

2)    Authors report a remarkably small footprint of the proposed device (Table 1) in comparison with other solutions. However, the TE-to-TM polarization conversion ratio is 96.5%. Is it possible to compare this value with performance of other devices listed in Table 1? Such a conversion ration implies that the output beam is elliptically polarized and that in order to obtain a “clean” TM polarized output wave one will need to add a polarizer to the device. At what extent this may increase the device footprint?
